# Extending Ensemble Kalman Filter Algorithms to Assimilate Observations with an Unknown Time Offset

Elia Gorokhovsky[1,2], Jeffrey L. Anderson[1]

[1]National Center for Atmospheric Research, Boulder, CO, USA

[2]Current affiliation California Institute of Technology, Pasadena, CA, USA

*Correspondence to*: Elia Gorokhovsky (eliapgorokhovsky@gmail.com), Jeffrey Anderson (jla@ucar.edu)

**Abstract.** Data assimilation (DA), the statistical combination of computer models with measurements, is applied in a variety of scientific fields involving forecasting of dynamical systems, most prominently in atmospheric and ocean sciences. The existence of misreported or unknown observation times (time error) poses a unique and interesting problem for DA. Mapping

observations to incorrect times causes bias in the prior state and affects assimilation. Algorithms that can improve the performance of ensemble Kalman filter DA in the presence of observing time error are described. Algorithms that can estimate the distribution of time error are also developed. These algorithms are then combined to produce extensions to ensemble Kalman filters that can both estimate and correct for observation time errors. A low-order dynamical system is used to evaluate the performance of these methods for a range of magnitudes of observation time error. The most successful algorithms must

explicitly account for the nonlinearity in the evolution of the prediction model.

## 1 Introduction

Ensemble data assimilation (DA) is one of the tools of choice for many earth system prediction applications including numerical weather prediction and ocean prediction. DA is also applied for a variety of other earth system applications like sea ice (Zhang et al., 2018), space weather (Chartier et al., 2014), pollution (Ma et al., 2019), paleoclimate (Amrhein, 2020), and

the earth's dynamo (Gwirtz et al., 2021). While DA was originally applied to generate initial conditions for weather prediction, it is also used for many related tasks like generating long-term reanalyses (Compo et al., 2011), estimating prediction model error (Zupanski and Zupanski, 2006), and evaluating the information content of existing or planned observing systems (Jones et al., 2014).

DA can also be used to explore other aspects of observations. An important part of many operational DA prediction systems is estimation and correction of the systematic errors (bias) associated with particular instruments (Dee and Uppala, 2009). Estimating the error variances, comprised of both instrument error and representativeness error (Satterfield et al., 2017), associated with particular observations is also possible (Desroziers et al., 2005) and can be crucial to improving the quality of DA products. DA methods have also been extended to explore problems with the forward operators, the functions used to

predict the value of observations given the state variables of the prediction model. These techniques can focus on particular aspects of forward operator deficiencies (Berry and Harlim, 2017) or attempt to do a more general diagnosis that can improve arbitrary functional estimates of forward operators, for instance an iterative method that can progressively improve the fit of the forward observation operator to the observations inside the data assimilation framework (Hamilton et al., 2019). Here, DA methods for estimating and correcting errors in the time associated with particular observations are explored.


Most observations of the earth system being taken now have precise times associated with them that are a part of the observation metadata. However, this is a relatively recent development for most applications. Even for the radiosonde network which is one of the foundational observing systems for the mature field of numerical weather prediction, precise time metadata has only been universally available for a few decades (Haimberger, 2007). Before the transition to current formats for encoding

and transmitting radiosonde observations, many radiosonde data did not include detailed information about ascent time or the time of observations at a particular height. Even the exact launch time was not always available in earlier radiosonde data that are a key part of atmospheric reanalyses for the third quarter of the 20th century (Laroche and Sarrazin, 2013).

This lack of time information is also a problem for surface-based observations, especially those taken before the radiosonde

era which relied on similarly limited encoding formats. Ascertaining the time of observations becomes increasingly problematic as one goes further into the past. As an example, coordinated time zones were not defined in the Unites States until the 1880s, resulting in local time uncertainty of minutes to hours in extreme cases. In fact, the major push for establishing coordinated time was motivated by the need for consistent atmospheric observing systems (Bartky, 1989). Similar issues were resolved earlier or later in other countries and not resolved globally until the 20th century.


As historic reanalyses extend further back in time (Slivinski et al., 2019), the lack of precise time information associated with observations can become an important issue. There is also a desire to use less quantitative observations taken by amateur observers and recorded in things like logbooks and diaries. An example is the assimilation of total cloud cover observations from personal records in Japan (Toride et al., 2017). While individual observers might have rigorous observing habits, the

precise time at which their observations were taken often remains obscure. Curiously, the problem of time error may be less for observations used for historical ocean reanalyses (Giese et al., 2016). This is because a precise knowledge of time was required for navigation purposes. Nevertheless, observations obtained from depth can involve unknown delays and failures to record the exact time associated with observations can remain (Abraham et al., 2013).

Even older observations, for instance those associated with paleoclimate, can have greater time uncertainty. Here, the fundamental relationship between the observations and the physical state of the climate system is poorly known and identifying the appropriate time scales is crucial to improved DA (Amrhein, 2020). Observations related to the evolution of the geosphere can have even more problematic time uncertainty. Initial work on using DA to reconstruct the evolution of the earth's

geodynamo highlight the problems associated with specifying the time that should be associated with various observations

(Gwirtz et al., 2021).

Failing to account for errors in the time associated with an observation can lead to significantly increased errors in DA results. This is especially true if time errors are correlated for a set of observations since they can result in consistently biased forward operators. Section 2 briefly describes the problem of observation time error while Sect. 3 discusses extensions to ensemble

DA algorithms that can explicitly use information about some aspects of time error. Section 4 describes several algorithmic extensions of ensemble DA that can provide estimates of time error distributions. Section 5 describes an idealized test problem while Sect. 6 presents algorithms combining the results of Sects. 3 and 4 to produce a hierarchy of ensemble DA algorithms that both estimate and correct for observation time error. Section 7 presents results of applying these algorithms and Sect. 8 includes discussion of these results and a summary.

**2 Statement of the problem**

The vector $\chi(t)$ is the time-dependent state of the dynamical system of interest and is defined at a set of discrete times $\{t_i\}, i = 0 \dots$ where $t_{i+1} = t_i + \Delta t$. The state $\chi$ is assumed to be observed at evenly spaced analysis times starting at 0 with a period of $P\Delta t$, $t_{k+1}^a = t_k^a + P\Delta t$, where $P$ is an integer. However, at each analysis time, the actual observation is taken at an observation time, $t_k^o = t_k^a + \varepsilon_k^t$ where the time offset $\varepsilon_k^t$ is unknown. In this paper, we make the simplifying assumption that $\varepsilon_k^t$ is drawn

from a normal distribution with mean $\mu_t$ and variance $\sigma_t^2$. In practice, including the experiments in this paper, only the case where $\mu_t$ is assumed to be 0 is relevant. This is because if $\mu_t$ is known, it is easy to reduce the problem to the case where $\mu_t = 0$ by advancing the ensemble by $-\mu_t$ when initializing it. On the other hand, if $\mu_t$ is unknown, then the problem is indistinguishable from the case where $\mu_t = 0$, but with an unknown bias in the initial ensemble. If the assimilation scheme is working properly, this bias should disappear over time anyway.


The time errors involved with many real measurements could be distinctly non-Gaussian. For instance, there is reason to believe clock errors may be skewed. For real application, it would be important to involve input from experts with detailed knowledge on the expected time error distributions. The case where time error is non-Gaussian can be approached using the same arguments as in Sect. 4, but is not explored further here.


The observations have an error $\boldsymbol{\varepsilon}_k^x \sim N(\mathbf{0}, \mathbf{R})$ with diagonal observation error covariance matrix $\mathbf{R}$. In this work, the observation operator is taken to be the identity, though this assumption is discussed further in Sect. 8. Hence, the observation at the $k$th analysis time is $\boldsymbol{y}_k^o \sim N(\boldsymbol{\chi}(t_k^o), \mathbf{R})$. When simulating this problem numerically, we do not have exact knowledge of $\boldsymbol{\chi}(t_k^o)$ when $t_k^o$ is not an integer multiple of $\Delta t$; linear interpolation is used to approximate $\boldsymbol{\chi}(t_k^o)$ in these cases. This

approximation uses the assumption that $\Delta t$ is small enough that $\chi(t)$ is nearly linear at the scale of one timestep; if this assumption is violated even basic ensemble Kalman filters can be ineffective.

## 3 Extending ensemble Kalman filters

Algorithms are described to extend ensemble Kalman filters (Burgers et al., 1998; Tippett et al., 2003) to use information about the time offset of observations. Suppose the time offset at the current analysis time, conditioned on the value of the current

observation, is distributed as $\varepsilon_k^t \mid y_k^o \sim N(\tilde{\mu}_{t,k}, \tilde{\sigma}_{t,k}^2)$. An ensemble Kalman filter assimilation for this analysis time can make use of this information by adjusting the prior ensemble estimate (the result of applying the forward operator to each ensemble state) and the observation error variance. First, the ensemble prior estimate of the observations, $y_n^p$, $n = 1 \dots N$, where subscript $n$ indexes the ensemble member and $N$ is the ensemble size, can be selected to simulate observations taken at (or near) time $t_k^a + \tilde{\mu}_{t,k}$; this time is the maximum likelihood estimate of the observation time. Second, the specified observation

error covariance matrix $\mathbf{R}$ can be augmented to include contributions from uncertainty due to the variance $\sigma_\tau^2$ of the estimate of the time offset.

Two methods of obtaining the prior mean and observation error variance are explored in the following subsections. In both cases, assume that an ensemble of prior estimates of the true state, $x_n^p(t_i)$ is available at the same discrete times $t_i$ as the truth

$\chi(t_i)$.

### 3.1 Extrapolation

Define $\boldsymbol{v}$ as the time derivative of the prior ensemble mean at the analysis time,

$$\boldsymbol{v} = \frac{\mathrm{d}\overline{x^p}}{\mathrm{dt}}(t_k^a), \tag{1}$$

where the overbar represents an ensemble mean. The prior state for each ensemble member can then be linearly extrapolated

to the most likely observation time,

$$\boldsymbol{y}_n^p = x_n^p(t_k^a) + \tilde{\mu}_{t,k}\boldsymbol{v} \tag{2}$$

(recall that the observation operator here is the identity.) This could also be done with additional cost by using the time derivative of each ensemble member, but that method is not explored here. The uncertainty in the time offset also leads to increased uncertainty in the observations. A linear approximation gives an enhanced observation error covariance matrix of

$\mathbf{E} = \mathbf{R} + \mathrm{diag}(\tilde{\sigma}_{t,k}^2\boldsymbol{v}^2). \tag{3}$

This approach assumes that time errors are small enough compared to the characteristic time scale of the system that the linearity approximation is valid. If this is not the case, it is more appropriate to use the more expensive method discussed in the next section. It also assumes that $\boldsymbol{v}$ is a good estimate of the time derivative of the truth itself, $\frac{d\boldsymbol{\chi}}{dt}(t_k^a)$.

### 3.2 Interpolation

A prior estimate of the observations for each ensemble can be obtained by linearly interpolating the values of the state to time $t_k^a + \tilde{\mu}_{t,k}$. It is convenient to require that this time is not earlier than the previous analysis time or later than the next analysis time, $t_{k-1}^a \le t_k^a + \tilde{\mu}_{t,k} \le t_{k+1}^a$. In order to interpolate to times between those limits, it is necessary to run the prior ensemble forecasts for up to twice as long as for a normal ensemble Kalman filter, out to $t_{k+1}^a$. This is no more than a doubling of the computation cost of prior forecasts for each analysis. The prior ensemble members must also be stored at all times between

the previous and next analysis times to facilitate interpolation. Computing the adjusted observation error variance in a more accurate way than just extrapolating (method 1 above) appears to be costly and complex and is not explored further here.

### 4 Computing estimates of the time offset

The previous section has presented algorithms to extend ensemble Kalman filters to cases where an estimate of the distribution of time offset $\tau$ at an analysis time is known. This section presents algorithms for estimating the distribution of $\tau$ at an analysis

time for use in an ensemble Kalman filter. Recall that $\mu_t$ is assumed to be 0 in Sect. 5 and onward.

### 4.1 No correction

The distribution is (incorrectly) assumed to be $\varepsilon_k^t \mid \boldsymbol{y}_k^o \sim N(0,0)$, so the default ensemble Kalman filter is applied.

### 4.2 Variance only

The distribution from which the time offset is drawn is used, $\varepsilon_k^t \mid \boldsymbol{y}_k^o \sim N(\mu_t, \sigma_t^2)$, without updating it based on the observation.

Using this with extrapolation results in no change to the prior mean but an increased observation error variance.

### 4.3 Impossible linear estimate

This algorithm assumes that the difference between the observation and the truth at the analysis time, $\tilde{\boldsymbol{d}} = \boldsymbol{y}_k^o - \boldsymbol{\chi}(t_k^a)$ is known; this is not possible in real systems where the truth is unknown (hence the name 'impossible') but provides an interesting baseline for practical algorithms. A method that drops this assumption (and is therefore possible) is discussed in the next

section.

Assuming the system has locally linear behavior near time $t_k^a$, if $\varepsilon_k^t$ is sufficiently small we can approximate $\chi(t_k^o) = \chi(t_k^a + \varepsilon_k^t)$ as $\chi(t_k^a) + \boldsymbol{v}\varepsilon_k^t$. In that case, the difference between the observation and the truth is approximately $\widetilde{\boldsymbol{d}} = (\boldsymbol{y}_k^o - \chi(t_k^o)) + (\chi(t_k^o) - \chi(t_k^a)) = \boldsymbol{\varepsilon}_k^x + \boldsymbol{v}\varepsilon_k^t$.


We want to find the relative likelihood of a particular time offset $\tau$, i.e., the relative PDF $p(\tau)$ of the distribution of $\varepsilon_k^t \mid \boldsymbol{y}_k^o$. By Bayes' theorem, recalling that $\widetilde{\boldsymbol{d}}$ is known, we have

$$p(\tau) \propto P\big(\varepsilon_k^t = \tau \mid \boldsymbol{\varepsilon}_k^x + \boldsymbol{v}\varepsilon_k^t = \widetilde{\boldsymbol{d}}\big) \propto P\big(\boldsymbol{\varepsilon}_k^x + \boldsymbol{v}\varepsilon_k^t = \widetilde{\boldsymbol{d}} \mid \varepsilon_k^t = \tau\big)P(\varepsilon_k^t = \tau). \tag{4}$$

$\boldsymbol{\varepsilon}_k^x + \boldsymbol{v}\varepsilon_k^t = \widetilde{\boldsymbol{d}}$ conditioned on $\varepsilon_k^t = \tau$ if and only if $\boldsymbol{\varepsilon}_k^x = \widetilde{\boldsymbol{d}} - \tau\boldsymbol{v}$. Since $\boldsymbol{\varepsilon}_k^x$ and $\varepsilon_k^t$ are independent, the assumption that $\boldsymbol{\varepsilon}_k^x$

is normally distributed (with mean $\boldsymbol{0}$ and covariance matrix $\mathbf{R}$) gives

$$P\big(\boldsymbol{\varepsilon}_k^x + \boldsymbol{v}\varepsilon_k^t = \widetilde{\boldsymbol{d}} \mid \varepsilon_k^t = \tau\big) = P\big(\boldsymbol{\varepsilon}_k^x = \widetilde{\boldsymbol{d}} - \tau\boldsymbol{v}\big) \propto \exp\left(-\tfrac{1}{2}(\widetilde{\boldsymbol{d}} - \tau\boldsymbol{v})^T \mathbf{R}^{-1}(\widetilde{\boldsymbol{d}} - \tau\boldsymbol{v})\right). \tag{5}$$

Finally, the assumption that $\varepsilon_k^t$ is also normally distributed (with mean $\mu_t$ and variance $\sigma_t^2$) gives

$$p(\tau) \propto \exp\left(-\tfrac{1}{2}(\widetilde{\boldsymbol{d}} - \tau\boldsymbol{v})^T \mathbf{R}^{-1}(\widetilde{\boldsymbol{d}} - \tau\boldsymbol{v})\right)\exp\left(-\tfrac{1}{2}(\tau - \mu_t)^2 \sigma_t^{-2}\right) \tag{6}$$

$$p(\tau) \propto \exp\left(-\tfrac{1}{2}\left[\widetilde{\boldsymbol{d}}^T \mathbf{R}^{-1}\widetilde{\boldsymbol{d}} + \mu_t^2 \sigma_t^{-2} - \tau\big(\widetilde{\boldsymbol{d}}^T \mathbf{R}^{-1}\boldsymbol{v} + \boldsymbol{v}^T \mathbf{R}^{-1}\widetilde{\boldsymbol{d}} + 2\mu_t\sigma_t^{-2}\big) + \tau^2\beta\right]\right), \tag{7}$$

where $\beta = \boldsymbol{v}^T \mathbf{R}^{-1}\boldsymbol{v} + \sigma_t^{-2}$. Note that since $\mathbf{R}$ is a covariance matrix, it is real symmetric (hence self-adjoint), so $\widetilde{\boldsymbol{d}}^T \mathbf{R}^{-1}\boldsymbol{v} = \boldsymbol{v}^T \mathbf{R}^{-1}\widetilde{\boldsymbol{d}}$:

$$p(\tau) \propto \exp\left(-\tfrac{\beta}{2}\left[\tau^2 - 2\tau\frac{\boldsymbol{v}^T \mathbf{R}^{-1}\widetilde{\boldsymbol{d}} + \mu_t\sigma_t^{-2}}{\beta} + \frac{\widetilde{\boldsymbol{d}}^T \mathbf{R}^{-1}\widetilde{\boldsymbol{d}} + \mu_t^2\sigma_t^{-2}}{\beta}\right]\right). \tag{8}$$

Since any constants may be absorbed into the proportionality, completing the square yields

$$p(\tau) \propto \exp\left(-\tfrac{\beta}{2}\left[\tau - \frac{\boldsymbol{v}^T \mathbf{R}^{-1}\widetilde{\boldsymbol{d}} + \mu_t\sigma_t^{-2}}{\beta}\right]^2\right). \tag{9}$$

This is the PDF of a normal with mean

$$\tilde{\mu}_{t,k} = \frac{\boldsymbol{v}^T \mathbf{R}^{-1}\widetilde{\boldsymbol{d}} + \mu_t\sigma_t^{-2}}{\beta} = \frac{\boldsymbol{v}^T \mathbf{R}^{-1}\widetilde{\boldsymbol{d}} + \mu_t\sigma_t^{-2}}{\boldsymbol{v}^T \mathbf{R}^{-1}\boldsymbol{v} + \sigma_t^{-2}} \tag{10}$$

and variance

$$\tilde{\sigma}_{t,k}^2 = \frac{1}{\beta} = \frac{1}{\boldsymbol{v}^T \mathbf{R}^{-1}\boldsymbol{v} + \sigma_t^{-2}}. \tag{11}$$

### 4.4 Possible linear estimate

In real applications, the difference between the observation and the truth at the analysis time cannot be computed, but the difference between the observation and the prior ensemble mean, $\boldsymbol{d} = \boldsymbol{y}_k^o - \overline{\boldsymbol{x}^p}(t_k^a)$ can. Linearly extrapolating (again,

assuming sufficiently linear local behavior near $t_k^a$) gives an estimate $\boldsymbol{d} \approx \boldsymbol{\varepsilon}_k^x + \boldsymbol{v}\varepsilon_k^t - \boldsymbol{\varepsilon}_k^p$, where $\boldsymbol{\varepsilon}_k^p \sim N(\boldsymbol{0}, \boldsymbol{\Sigma}^p(t_a^k))$ is a draw from the prior ensemble sample covariance at the analysis time. Here $\boldsymbol{\Sigma}^p(t_a^k)$ refers to the sample covariance matrix of the prior ensemble at the analysis time $t_a^k$. This assumes that the prior ensemble distribution is consistent with the truth so that the truth over many analysis times is statistically indistinguishable from the prior ensemble members. In real applications, this is never the case. For instance, any practical problem would certainly have model deficiencies so that the prior would be biased and $\boldsymbol{\varepsilon}_k^p$ would have a non-zero mean.

Defining the difference of the observation error $\boldsymbol{\varepsilon}_k^x$ and the prior uncertainty $\boldsymbol{\varepsilon}_k^p$ as $\boldsymbol{\varepsilon}_k^\delta = \boldsymbol{\varepsilon}_k^x - \boldsymbol{\varepsilon}_k^p$, we have $\boldsymbol{\varepsilon}_k^\delta \sim N(\boldsymbol{0}, \mathbf{R} + \boldsymbol{\Sigma}^p(t_k^a))$. The analysis for the impossible linear estimate can be repeated by solving for the probability

$$p(\tau) = P\big( \varepsilon_k^t = \tau \mid \boldsymbol{\varepsilon}_k^\delta + \boldsymbol{v}\varepsilon_k^t = \boldsymbol{d} \big). \tag{12}$$

in the same fashion. The result is that

$$\tilde{\mu}_{t,k} = \frac{\boldsymbol{v}^T[\mathbf{R}+\boldsymbol{\Sigma}^p(t_k^a)]^{-1}\,\boldsymbol{d} + \mu_t\sigma_t^{-2}}{\boldsymbol{v}^T[\mathbf{R}+\boldsymbol{\Sigma}^p(t_k^a)]^{-1}\boldsymbol{v} + \sigma_t^{-2}} \tag{13}$$

and

$$\sigma_\tau^2 = \frac{1}{\boldsymbol{v}^T[\mathbf{R}+\boldsymbol{\Sigma}^p(t_k^a)]^{-1}\boldsymbol{v} + \sigma_t^{-2}} \tag{14}$$

Under the linearity assumption, because time error contributes a Gaussian error $\boldsymbol{v}\varepsilon_k^t$ to the observation, it is statistically difficult to distinguish between the usual observation error and error due to time offset. This can lead to time error estimates with a magnitude that is too large. This error can propagate to subsequent analysis times and lead to biased prior estimates that can result in unstable feedback in the assimilation. Sect. 6.3 presents evidence of this problem and describes a solution that works for the test applications explored there.

## 4.5 Nonlinear estimate

As for the interpolation method in Sect. 3.2, assume that an ensemble of prior estimates of the true state, $\boldsymbol{x}_n^p(t_i)$ is available at the same discrete times as the truth for $t_{k-1}^a \leq t_i \leq t_{k+1}^a$. Assume that the prior is normal with the ensemble giving a good estimate of the prior distribution, i.e., a priori, the true state at time $t_i$, $\boldsymbol{\chi}(t_i)$ is drawn from the multivariate normal distribution with mean $\overline{\boldsymbol{x}^p}(t_i)$ (the average of the prior ensemble at time $t_i$) and covariance $\boldsymbol{\Sigma}^p(t_i)$ (the covariance of the prior ensemble). Recall that the observation error $\varepsilon_k^x$ is assumed to be drawn from a normal distribution with mean $\boldsymbol{0}$ and covariance $\mathbf{R}$. Hence, conditioned on $t_k^o = t_i$ for some $t_i$, the *relative* likelihood of making the observation $\boldsymbol{y}_k^o = \boldsymbol{y}$ for some $\boldsymbol{y}$ is given by a sum of Gaussians:

$$P(\boldsymbol{y}_k^o = \boldsymbol{y} \mid t_k^o = t_i) = P(\varepsilon_k^\delta = \boldsymbol{y} - \boldsymbol{y}_k^o \mid t_k^o = t_i) \propto N(\overline{\boldsymbol{x}^p}(t_i), \boldsymbol{\Sigma}^p(t_i) + \mathbf{R}; \boldsymbol{y}), \tag{15}$$

200 where $N(\boldsymbol{\mu}, \boldsymbol{\Sigma}; \boldsymbol{x})$ refers to the PDF of a normal distribution with mean $\boldsymbol{\mu}$ and covariance matrix $\boldsymbol{\Sigma}$, evaluated at $\boldsymbol{x}$ (in one dimension, the PDF of a normal distribution with specified mean and variance, evaluated at a point).

On the other hand, recall that the relative likelihood that $t_k^o = t_i$ is a normal distribution with mean $t_k^a$ and variance $\sigma_t^2$. Hence, Bayes' theorem gives that for each $t_i \in [t_{k-1}^a, t_{k+1}^a]$ , the relative likelihood that the offset observation was taken at this time

205 is given by the following product:

$$P(t_k^o = t_i) \propto N(\overline{\boldsymbol{x}^p}(t_i), \boldsymbol{\Sigma}^p(t_i) + \mathbf{R}; \boldsymbol{y}_k^o) \, N(t_k^a + \mu_t, \sigma_t^2; t_i), \tag{16}$$

The value of $t_i$ with the largest relative likelihood given by Eq. (16) is assumed to correspond to the maximum likelihood estimate of the time offset, $\tilde{\mu}_{t,k} = t_k^a - t_i$. It is complex and expensive to compute a nonlinear estimate of the variance of the offset, $\tilde{\sigma}_{t,k}^2$, and that is not explored here. It is also possible to compute the $\tilde{\mu}_{t,k}$ in other related ways. For example, the

210 likelihood weighted average of the $\{t_i\}$, could be used instead. This was found to make only small differences to the results described in Sect. 7.

## 5 Low-order model test problems

A set of assimilation methods described in the next section are applied to the 40-variable model described in Lorenz and Emmanuel (1998), referred to as the L96 model. The model has 40 state variables $X_1, \dots, X_{40}$ (with $X_{40}$ also labelled $X_0$ and

215 $X_{39}$ also labelled $X_{-1}$), and the evolution of the model is given by the following 40 differential equations:

$$\frac{dX_i}{dt} = X_{i-1}(X_{i+1} - X_{i-2}) - X_i + F, \qquad\qquad i = 1, \dots, 40. \tag{17}$$

The forcing parameter $F$ is set to 8 in this work. This value was chosen by Lorenz and Emmanuel (1998) for their baseline exploration because it is one of the smallest values that results in chaotic dynamics. This value is used in a large number of applications of the L96 model (for example Anderson 2001; Dirren and Hakim, 2005; van Leeuwen 2010).


 A fourth-order Runge-Kutta time differencing scheme is applied with a non-dimensional timestep of $\Delta t = 0.01$ instead of the 0.05 that is more frequently used in previous work. The choice to use a smaller timestep is intended to make the timestep smaller than the values of $\sigma_t$ for which the algorithm was tested. If $\Delta t$ were much larger than $\sigma_t$, then most true observation times would be within one timestep of the reported observation times. Since linear interpolation was used to compute the states

225 of the system between timesteps, this would lead to time error contributing a linear factor to overall observation error. In practical applications, we are quite interested in the effect of the system's nonlinearity on the total error in the presence of time error, which would not be represented in the experiment if the timestep were larger.

Results are explored for 5 different simulated observing systems that differ by the analysis period, $P$, with which observations

230 are supposed to be taken. The periods are 5, 10, 15, 30 and 60 timesteps corresponding to 0.05, 0.1, 0.15, 0.3 and 0.6 time

units. Each experiment performs 1100 analysis steps and the first 100 analysis steps (corresponding to between 500 and 6000 timesteps) are always discarded. Inspection of time series of prior ensemble mean error suggests that the system is equilibrated well before 100 steps for all experiments.

For a given analysis period, the L96 model is integrated from an initial condition of 1.0 for the first state variable and zero for all others to generate truth trajectories. Eleven initial conditions are generated by saving the state every 1100 analysis times. The first initial condition is used to empirically tune localization and inflation, and the other ten are used for ten trials using the tuned values.

For each observing system, several values of the standard deviation of the observation time offset $\sigma_t$ are explored. The combination of $\sigma_t$ and an analysis period $P$ define a case. Table 1 shows the cases explored. For each method applied to each case, a set of 49 assimilation experiments is performed using pairs of Gaspari-Cohn (Gaspari and Cohn 1999) localization *half-widths* selected from the set {0.125, 0.15, 0.175, 0.2, 0.25, 0.4, ∞} and fixed multiplicative *variance* inflation (Anderson and Anderson 1999) selected from the set {1, 1.02, 1.04, 1.08, 1.16, 1.32, 1.64}. The pair of half-width and inflation that
produces the minimum posterior ensemble mean root mean square error from the truth is used for ten subsequent experiments for the case that differ only in the initial truth condition.

   At each analysis time $k$, all 40 state variables are observed at a time $t_k^o$ that has an offset $\varepsilon_k^t$ from the analysis time, $t_k^o = t_k^a + \varepsilon_k^t$. All observations at a given time share the same time offset which is generated as a random draw from a truncated normal
distribution with mean 0, variance $\sigma_t^2$ and bounds at $\pm P\Delta t$.

**Table 1**: List of observing system cases explored. For each of five analysis periods, a number of different values for the time offset standard deviation were explored.

| Analysis Period | $\sigma_t$ |
|---|---|
| 0.05 | 0, 0.0125, 0.025, 0.05 |
| 0.10 | 0, 0.0125, 0.025, 0.05, 0.1 |
| 0.15 | 0, 0.0125, 0.025, 0.05, 0.1 |
| 0.30 | 0, 0.0125, 0.025, 0.05, 0.1, 0.2 |
| 0.60 | 0, 0.0125, 0.025, 0.05, 0.1, 0.2 |

   Figure 1 shows a short segment of the trajectory of the truth, $\chi(t_i)$, for a single L96 state variable and the generation of
observations for the case with analysis period 0.60 and time error standard deviation 0.2. The true observation values $y_k^{tr}$ for

each state variable are generated by linearly interpolating the true state trajectory $\chi(t_i)$ that is available every 0.01 time units to the offset observing times $t_k^o$ (blue circles in Fig. 1). The observation error variance is 1 for all experiments and the actual observations that are assimilated, $\mathbf{y}_k^o$ (yellow 'X' in Fig. 1) are generated by adding an independent draw from $N(0,1)$ to $\mathbf{y}_k^{tr}$ for each of the 40 state variables.


Figure 1 also shows the value of the state linearly extrapolated from the analysis time to the observed time as a blue vector and teal '+'. This is analogous to the extrapolation performed by Eq. (2) using the ensemble mean estimate as a base point. In Eq. (2), all ensemble members are shifted by the same vector analogous to the blue one in Fig. 1, but that vector is an approximation of the one shown in Fig. 1. The figure also gives a feeling for how nonlinear the time offset problem is at a particular analysis

time, i.e., where the linearity assumptions in Eq. (2) would fail. For example, near time 369.5, the linearity assumption fails and leads to additional error, while near 371.5 it is partially effective and gives a better estimate for the truth. In cases where the linearity assumption is likely to fail, it may be more appropriate to use the interpolation method discussed in Sect. 3.2.

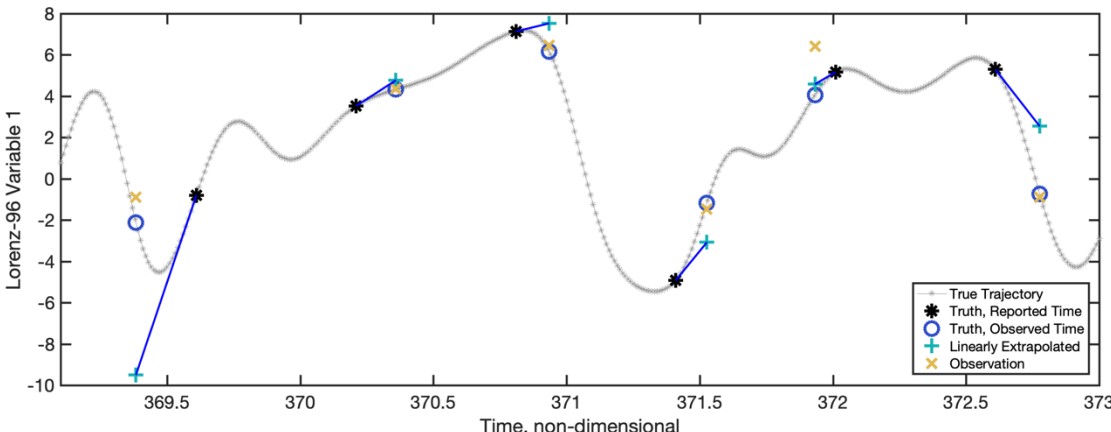


**Figure 1**: A short segment of the truth for a state variable and the observation generation process from the case with analysis period 0.6 and time error standard deviation 0.2. The true trajectory is indicated by the small grey asterisks every 0.01 time units. The black asterisks indicate the true value at each analysis time. The blue circles are the truth at the actual observed time (the analysis time plus the observation time offset for that analysis time). The yellow crosses are the actual observations that

are assimilated and are generated by adding a random draw from $N(0,1)$ to the truth at the actual observed time. The teal '+' indicate the result of linearly extrapolating the truth at the analysis time to the actual observed time using the time derivative of the model at the analysis time ($\mathbf{v}$ in Eq. (1)); a blue line segment connects the truth to the extrapolated value.

## 6 Assimilation Methods

Five assimilation methods were tested for each L96 case. All applied a standard ensemble adjustment Kalman filter (EAKF, Anderson 2001) with 80 members using a serial implementation (Anderson 2003) to update the ensemble with observations. All but the first method made adjustments to the prior observation ensemble and/or the observation error covariance to deal with the observation time offset.

### 6.1 No correction (referred to as NOCORRECTION)

Observations were assimilated with a standard EAKF. This is consistent with the assumption made in Sect. 4.1 that the time offset is $\varepsilon_k^t \mid \boldsymbol{y}_k^o = 0$.

### 6.2 Adjust observation error variance only (referred to as VARONLY)

This method assumed time offset $\varepsilon_k^t \mid \boldsymbol{y}_k^o \sim N(0, \sigma_t^2)$ as in Sect. 4.2 and only adjusted the observation error variance using the linear approximation given in Eq. (3).

### 6.3 Possible linear correction (referred to as LINEAR)

This method used Eq. (13) and Eq. (14) to compute the mean and variance of the time offset. This distribution for $\tau$ was then
used with the extrapolation method of Sect. 3.1, using Eq. (2) to compute prior ensemble estimates of each observation and Eq. (3) to compute the observation error variance.

A naive application of this method was not successful in any of the L96 cases. The tuned assimilations worked successfully for some number of analysis times, but the RMSE of the ensemble mean always began to increase with time before 1100
analysis times and results were worse than for NOCORRECTION. The magnitude of the estimate of the mean value of the time offset $\left|\tilde{\mu}_{t,k}\right|$ would also systematically increase with time.

This occurred because of the statistical challenge of separating observation time offset from prior model error. Suppose this method was applied to a model with only a single time varying variable that is observed. The prior ensemble mean will almost
always have an error. If, for example, that error has the same sign as the time tendency of the model at the analysis time, the linear correction method will attribute part of that error to a time offset in the observation and will not correct the error as strongly as it would if no time offset were assumed. This means that the forecast at the next analysis time is likely to be consistent with the model state at a time later than the analysis time. Again, the algorithm will attribute some of this error to a time offset in the observation. The net result is that the estimated model state is likely to drift further and further ahead of the
true trajectory in time.

To avoid this problem, estimates of the time offset that were (nearly) independent of the error for a given state variable were needed. This was accomplished by using a modified version of Eq. (13) to compute a separate value of $\tilde{\mu}_{t,k}$ for each observation:

$$\tilde{\mu}_{t,k}^{\approx m} = \frac{v^T[\mathbf{R}+\Sigma^p(t_k^a)]^{-1}d^{\approx m}}{v^T[\mathbf{R}+\Sigma^p(t_k^a)]^{-1}v+\sigma_t^{-2}}. \qquad m = 1 \ldots M \tag{18}$$

where $M$ is the number of observations (here, $M = 40$, the size of the model, for all experiments). The vector $d^{\approx m}$ is a modification of the original vector $d$, the distance between the observations and the prior ensemble mean at the analysis time. The $i$th component of $d^{\approx m}$ is given by

$$d_i^{\approx m} = \begin{cases} d_i, & \|i,m\| > T; \\ 0, & \|i,m\| \le T, \end{cases} \tag{19}$$

where $T$ is an integer cutoff threshold and $\|i,m\|$ is the cyclical distance in units of grid intervals between two variables in the 40-variable L96 model,

$$\|i,m\| = \begin{cases} |i-m|, & |i-m| \le 20; \\ 40-|i-m|, & |i-m| > 20. \end{cases} \tag{20}$$

For example, if $m = 35$, then $\|i,m\| \le 10$ for $25 \le i \le 40$ and $i \le 5$.

A subset of the components of the vector $d$ that correspond to observed state variables close to the $m$th state variable were set to 0, effectively eliminating the impact of these state variables on the estimated time offset for the $m$th observation. All results shown here for the LINEAR method used a threshold $T$ of 10 so that 21 components (out of 40) were set to zero. Larger or smaller values of $T$ increased the RMSE in tuning experiments performed for the case with analysis period $P = 0.3$ and time offset standard deviation $\sigma_t = 0.1$. It is likely that improved performance for other cases could result from retuning $T$, but this

was not explored here. Any applications of this algorithm to real problems would require tuning of the threshold.

### 6.4 Impossible linear correction (referred to as IMPOSSIBLE)

This method used Eq. (10) and Eq. (11) to compute the mean and variance of the time offset. This distribution for $\tau$ was then used with the extrapolation method of Sect. 3.1, using Eq. (2) to compute prior ensemble estimates of each observation and

Eq. (3) to compute the observation error variance. As noted, computing $\tilde{d}$ for use in Eq. (10) requires knowledge of the true state so this is not a practical algorithm. Knowledge of the truth prevents the drift away from the truth that necessitated the use of Eq. (16) for LINEAR.

### 6.5 Nonlinear correction (referred to as NONLINEAR)

The nonlinear algorithm in Sect. 4.5 was used to estimate the most likely value of the time offset $\tilde{\mu}_{t,k}$ and the interpolation method in Sect. 3.2 was used to adjust the prior estimates of each observation.

In addition to estimating the model state, each of the 5 methods also estimated the value of the time offset, $\tilde{\mu}_{t,k}$ at each analysis times. Methods NOCORRECTION, VARONLY and LINEAR used the possible estimate from Eq. (13), IMPOSSIBLE used the estimate from Eq. (10), and NONLINEAR used the estimate from Sect. 4.5.

## 7 Results

Figure 2 shows the results for the five methods applied to all cases. For each case, the RMSE of the prior ensemble mean is plotted for each of the ten trials made with each assimilation method. The results for different methods are distinguished by the color of the markers and the horizontal offset of the plot columns. Note that ranges of both axes vary across the figures and that the horizontal axis is logarithmic (with the exception of the value for no time offset).

The blue markers (leftmost) are the results of the NOCORRECTION method which ignores the time offset and does a standard ensemble adjustment Kalman filter using $N(0, 1)$ as the observation error. The VARONLY method, shown in orange (middle), accounts for the added uncertainty in the observation values due to the unknown time offset. VARONLY is better than NOCORRECTION for longer analysis periods and larger time error standard deviations. There are no cases for which VARONLY is obviously worse than NOCORRECTION.

The LINEAR method is shown in teal (second from left). For almost all cases, it generally produces smaller RMSE than NOCORRECTION with the relative improvement being largest for analysis period 0.1 and 0.15 and larger time error standard deviation. LINEAR produces larger RMSE than NOCORRECTION for all cases with analysis period 0.6. The poor performance for the cases with time error standard deviation greater than 0.15 is due to errors in the linear tangent approximation for the evolution of the L96 state trajectories (see examples for the 0.6 analysis period in Fig. 1). LINEAR applies the same increment to the observational error variance as VARONLY. It performs better than VARONLY for most cases. However, VARONLY is better than LINEAR for cases with period 0.6, showing that the additional linear correction to the prior ensemble is clearly inappropriate for these cases.

Additional insight into the performance of LINEAR can be gained from the results for IMPOSSIBLE, shown in black (second from right) in the figure. Not surprisingly, since it has access to the truth when estimating the offset, it always produces smaller RMSE than LINEAR (except for cases with no time error). For the 0.05, 0.1 and 0.15 analysis period cases, the RMSE for IMPOSSIBLE is nearly independent of the time error standard deviation. This is not the case for analysis periods 0.3 and 0.6 where the error increases as the time error standard deviation increases. The cause of this error increase is that the linear tangent approximation becomes inaccurate as the time error increases. However, especially for analysis period 0.6, IMPOSSIBLE does not produce significantly better RMSE than NOCORRECTION, even for smaller time error standard deviation where the

linear tangent approximation should normally be accurate. Apparently, the larger prior error resulting from infrequent observations dominates the errors introduced by the time error in these cases.

The NONLINEAR method plotted in yellow (rightmost) has additional information about the distribution of the time offset and almost always performs significantly better than NOCORRECTION. The relative importance of nonlinearity in the prior truth trajectories is revealed by comparing the RMSE for IMPOSSIBLE and NONLINEAR. For time error standard deviations smaller than 0.1, IMPOSSIBLE is almost always significantly better, but for time error standard deviation of 0.1 and 0.2, NONLINEAR is always better.

All methods also produce an estimate, $\tilde{\mu}_{t,k}$, of the true time offset, $\varepsilon_k^t$, at each analysis time, $t_k^a$. The RMSE of the estimate for each method for cases with analysis periods 0.1 and 0.3 are shown in Fig. 3 with the same color/position scheme as in Fig. 2. For NOCORRECTION, the offset is estimated using Eq. (13) even though the offset is not used in the algorithm. For LINEAR, the estimate is the estimate using all state variables from Eq. (13), not the revised estimates using Eq. (16). For IMPOSSIBLE, the offset is computed using Eq. (10).

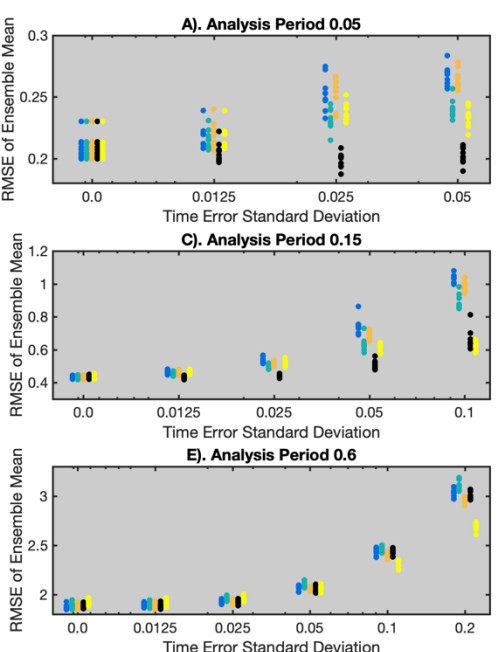

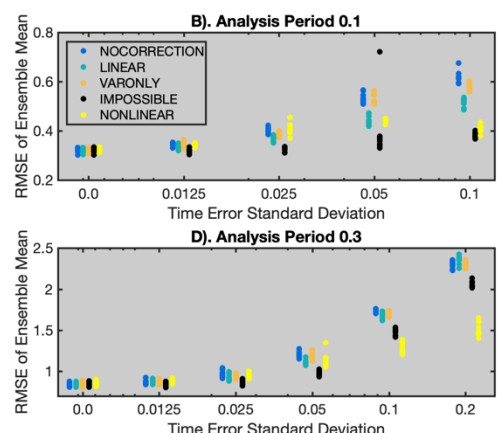

**Figure 2**: RMSE of the ensemble mean over 1000 analysis time steps for cases with analysis period (A) 0.05, (B) 0.1, (C) 0.15, (D) 0.3, or (E) 0.6 time units. Each dot in the graphs corresponds to an experiment run with a particular method, analysis period, and time error. Ten experiments were run for each method-analysis period combination. The horizontal axis is logarithmic except for the 0 value.

For analysis period of 0.1 (Fig. 3A), the estimates from all methods are always less than the specified time error standard deviation and become smaller fractions of the specified value as the value increases. This is because it is easier to detect time error when that error is relatively larger compared to the observation error. LINEAR and VARONLY have smaller RMSE

than NOCORRECTION for larger time error standard deviations with LINEAR being slightly better than VARONLY. The RMSE for NONLINEAR is much larger than for NOCORRECTION for smaller time error standard deviations. This is because the possible offset estimates are selected from the discrete set of times for which the truth and prior ensemble are computed (see Eq. (15)) which are spaced 0.01 time units apart. The time offset estimates for all other methods can take on any real value. For the case with time error standard deviation 0.1, the nonlinearity is large enough that the NONLINEAR estimate of

the offset is comparable to that produced by VARONLY and is better than NOCORRECTION.

For larger analysis period of 0.3 (Fig. 3B), the estimate from LINEAR is not better than NOCORRECTION, while VARONLY is better than LINEAR for larger time error standard deviations. In this case, NONLINEAR still has the largest RMSE for cases with time error standard deviation of 0.025 and 0.05, but has by far the smallest RMSE for cases with 0.1 and 0.2.


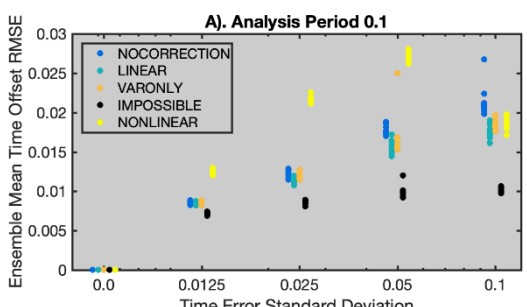 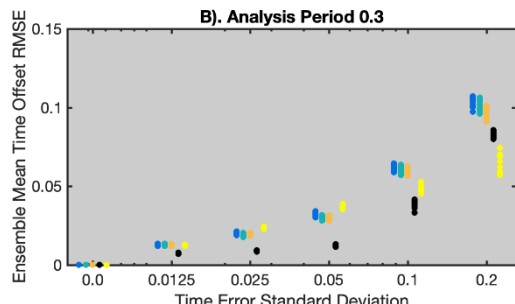

**Figure 3**: The RMSE of the estimate of the time offset for cases with analysis period (A) 0.1 time units and (B) 0.3 time units. Ten experiments were run for each of 5 methods with each method indicated by a different color. The horizontal axis is logarithmic except for the 0 value.

**8 Discussion and summary**

A number of simplifying assumptions were made in the algorithms described here. These include assuming that every state variable is observed directly, that all observations share the same time offset, that the observation error covariance matrix **R** is diagonal, and that the time offset variance, $\sigma_t^2$, is known a priori. Additionally, the assumption of linearity and the assumption that the average time offset, $\mu_t$, is 0 are discussed above.


It is straightforward to deal with some of these issues. The assimilation problem can be recast in terms of a joint phase space, where an extended model state vector is defined as the union of the model state variables and prior estimates of all observations

(Anderson 2003). Then, all observed quantities are model state variables by definition. However, for methods that use linear extrapolation via Eq. (1), the model equations are no longer sufficient. One can either develop equations for the time tendency of observations, or simply use finite difference approximations to compute $\nu$ for the extended state. It is even more straightforward to extend the method to cases where not all (extended) state variables are observed. Both the methods for using (Sect. 3) and computing (Sect. 4) information about the time offset at the current analysis time can be applied just to the variables that are observed.

Since a serial ensemble filter is being used for the actual assimilation, it is possible to partition the observations into subsets that are themselves assimilated serially. All observations that share a time offset can be assimilated as a subset, including a subset for those observations with no time offset.

All of the methods for estimating the offset at a given analysis time except NOCORRECTION make explicit use of $\sigma_t^2$, the variance of the distribution from which the offsets are drawn. If this is not known accurately, the performance of all the algorithms is expected to degrade. However, tests in which the value used in the assimilation was either 4 or 16 times larger than the actual value of $\sigma_t^2$ led to only limited increases in the RMSE of the various methods. It is also possible to refine the estimate of this variance by starting with a large value and examining the estimated values of the time offset that result.

The methods also assume that the observation error covariance matrix **R** is diagonal, which simplifies the derivation of the equations in Sects. 4.3 and 4.4 and allows the serial implementation. Removing this simplifying assumption requires computing and inverting matrices of size $M \times M$, where $M$ is the number of observations with mutually correlated errors. The increase in cost is the same as for algorithms that do not estimate time offset.

The methods described have a range of computational costs. The VARONLY method only requires a single evaluation of Eqs. (2) and (3) at each analysis time step and has an incremental cost that is a tiny fraction of the NOCORRECTION base filter. The LINEAR method requires an evaluation of Eq. (16) for every observation, and Eq. (16) requires the computation, storage, and inversion of a prior ensemble covariance matrix. However, this matrix could be reduced in size to only include the subset of observations that is used to compute the offset for each observation. It would be application specific to determine this size. For example, a radiosonde would make a large number of observations, e.g., temperature and wind at a number of levels, but many of these are correlated in time. We can capture most of the information about time offset from a smaller subset of the observations, and just do the inversion on those to make the matrix small compared to the total model size.

The NONLINEAR method involves a large amount of additional computation. The prior ensemble needs to be available over a range of times covering the possible offsets. In the idealized cases here, that meant that ensemble forecasts were required to extend to the second analysis time in the future, doubling the forecast model cost. Then Eq. (15) must be evaluated for each of

the available times. The dominant cost in Eq. (15) is computing the prior covariance matrix for the observations that share an offset. This requires $O(M^2)$ computations, where $M$ is the number of observations. Again, the relative cost would be highly application specific, but this method is the most expensive of the five.


The importance of accounting for observation time errors in many earth system DA applications remains unexplored. The range of methods discussed here have varying cost, but all could be applied for at least short tests in any application for which ensemble DA is already applicable. In particular, applications to atmospheric reanalyses for periods well before the radiosonde era seem to be especially good candidates for improvement. Future work will assess the algorithms presented here in both
observing system simulation and real observation experiments with global atmospheric models and observing networks from previous centuries.

**Code and data availability**

All code used to generate the data for this study, the generated data, code for creating the figures and figure files are available at https://github.com/jlaucar/npg_gorokhovsky

**Author contributions**

EG proposed the study, developed the mathematical methods, and implemented initial test code. JA solved the problems with the possible linear method, ran final tests and implemented figures. Both wrote the final report.

**Competing interests**

The authors declare they have no competing interests.

**Acknowledgements**

The authors would like to thank Peter Teasdale and Valerie Keeney of Summit Middle School and Paul Strode and Emily Silverman of Fairview High School for their commitment to science education and providing the authors with an opportunity to collaborate. The authors are indebted to NCAR's Data Assimilation Research Section team for providing guidance in developing the software used for this work.

**Financial support**

This material is based upon work supported by the National Center for Atmospheric Research, which is a major facility sponsored by the National Science Foundation under Cooperative Agreement No. 1852977.

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
