# Peer review of "Extending Ensemble Kalman Filter Algorithms to Assimilate Observations with an Unknown Time Offset"

_EGUsphere, 2022_

## Referee Comment (RC1)

Review of the manuscript 'Extending Ensemble Kalman Filter Algorithms to Assimilate Observations with an Unknown Time Offset' by Elia Gorokhovsky and Jeffrey L. Anderson.

The manuscript explores the impact of observation time errors in data assimilation (DA) experiments, using the EKF method. Different approaches of taking into account time errors are explored and compared in a systematic way. The organization of the manuscript (introduction, methods, results, conclusions) is generally good but may be improved in many places. The recommendation is the acceptance of the manuscript after the correction/explanation of some issues, presented bellow.

1) Line 19-21 Authors must include recent references of DA applications in the subjects referred in the manuscript: sea ice, space weather, pollution, paleoclimate, earth's dynamo. long-term reanalyses, estimating prediction model error and evaluating the information content of existing or planned observing systems.

2) Line 29 The phrase: 'attempt to do a completely general diagnosis (Hamilton et al., 2019)' is quite vague. Please rephrase explaining in more detail.

3) Line 76 The time offset or time error $\epsilon_k^t$ is assumed to have an unbiased, symmetric, Gaussian distribution. Give arguments to assume those hypotheses. For example, clock errors are much likely to be skewed due to time delays of several origins (electronics, propagation etc.).

4) Line 77 Instead of 'observation error covariance $\mathbf{R}$ diagonal' say 'diagonal observation error covariance $\mathbf{R}$'. In every reference to R, the word matrix must be included.

5) Line 90 (Eq 1). The state vector is represented by letter x(t) while before that is represented by the Greek letter χ(t). The notation must be consistent, all over the manuscript.

6) Eq 2 is assuming a direct observation vector y of the state vector x, i.e. an identity observation operator corresponding to the identity operator. The authors must say that explicitly in order to well understand the difference between x and y. Moreover in eq. 2 and 3, a linear correction of the state vector due to time errors is assumed. Justify this approximation. The linearity hypothesis is only valid for small enough time errors or slow enough dynamics (small time derivative of the state vector). The smallness of time errors is not guaranteed for every type of observations, for instance when observation time is quite fuzzy (e.g. qualitative observations issued from old data, logbooks and diaries).

7) Eq. 6 Use the consistent notation $\tau^2$ instead of $\tau$^2.

8) Explain the title of section 4.3 'Impossible linear estimate' in this section, not in the following one.

9) Line 140. Define the matrix $\Sigma$ within the text.

10) Line 150 In eq. 13 the vector d must be used instead of d-tilde.

11) Line 153. Authors say, 'It is statistically difficult to distinguish between observation error and error due to time offset' Again this is only valid when linearity hypothesis is assumed leading to the additivity of observation error variance with the error variance coming from the time offset. Say that in the text.

12) Line 170. Present the reason for the choice of the '40-variable Lorenz model'. It is worth to present the model equations here:

$dX_i\, dt = X_{i-1}(X_{i+1} - X_i - 2) - X_i + F$; $i = 1,...,40$, $X_0 \equiv X_{40}$, $X_{-1} \equiv X_{39}$, where the index is the positional index in a periodic ring.

Justify the use of parameter F=8, maybe due to the existence of a chaotic attractor. Present other works using the same model in DA experiments:

*Evenson, G. and Fario, N.: Solving for the generalized inverse of the Lorenz model, J. Meteorol. Soc. Jpn., 75, 229–243, 1997.*

*van Leeuwen, P. J.: Nonlinear data assimilation in geosciences: an extremely efficient particle filter, Q. J. Roy. Meteor. Soc., 136, 1991–1999, 2010*

and nonlinear statistics like:

*Pires, C.A.L. & Perdigão, R.A.P. (2015) Non-Gaussian interaction information: estimation, optimization and diagnostic application of triadic wave resonance. Nonlinear Processes in Geophysics, 22, 87-108. DOI:10.5194/npg-22-87-2015*

13) Line 192. The random draws of the time offset are not allowed to be larger than P$\Delta$t in absolute value. In that case, authors halve artificially the error to 1/2P$\Delta$t which cuts the occurrence of large time offsets. The most correct procedure would be to take a truncated normal distribution (https://en.wikipedia.org/wiki/Truncated_normal_distribution). Authors must comment on that.

14) Line 205. Authors say: 'This is analogous to the extrapolation performed by Eq. (2) using the ensemble mean estimate as a base point and gives a feeling for how nonlinear the time offset problem is at a particular analysis time.' Explain that describing the behavior in particular times (e.g. near time=369.6 the tangent linear hypothesis has completely failed).

15) Caption of Figure 1 must be completely revised. Refereed symbols do not correspond to the Table inserted in Figure 1 (e.g. green symbols do not exist).

16) In eqs. 17 and 18 do not use the right curly bracket }, since, mathematically, {} is used for the description of a set of members.

17) Lines 255-259. The definition of $d^m_i$ is quite confusing. Accordingly to the text, m is the positional index of the state vector. In lines 190-191, one says that all state variables are observed. Therefore, the observation vector has the same dimension than state vector and therefore the index i in $d^m_i$ (The subscript $i$ references the vector component for a given scalar observation) is quite confusing. Explain that much better, using different indices for different purposes.

18) Lines 260-265. The choice of the threshold T appears to be quite artificial and not generalizable to a generic model.

19) In the caption of Fig. 2, say that the RMSE in each of the 10 experiments (IC trials), correspond to a dot in the graphs, using a particular DA method (color), analysis period and time offset std.

---

## Author Comment (AC1)

**Reviewer 1:**

1. Line 19-21 Authors must include recent references of DA applications in the subjects referred in the manuscript: sea ice, space weather, pollution, paleoclimate, earth's dynamo. long-term reanalyses, estimating prediction model error and evaluating the information content of existing or planned observing systems

References have been added for each of these.

2. Line 29 The phrase: 'attempt to do a completely general diagnosis (Hamilton et al., 2019)' is quite vague. Please rephrase explaining in more detail.

Added text with a bit more detail, "do a more general diagnosis that can improve arbitrary functional estimates of forward operators, for instance an iterative method that can progressively improve the fit of the forward observation operator to the observations inside the data assimilation framework (Hamilton et al., 2019) "

3. Line 76 The time offset or time error $\varepsilon_t^k$ is assumed to have an unbiased, symmetric, Gaussian distribution. Give arguments to assume those hypotheses. For example, clock errors are much likely to be skewed due to time delays of several origins (electronics, propagation etc.).

The reviewer is certainly correct. We added a discussion related to the assumption that the error is unbiased and revised the equations in section 4 to deal with this case. We also added text that notes this challenge and mentions that expert knowledge (that the authors lack) could help with solutions.

"The time errors involved with many real measurements could be distinctly non-Gaussian. For instance, there is reason to believe clock errors may be skewed. For real application, it would be important to involve input from experts with detailed knowledge on the expected time error distributions. The case where time error is non-Gaussian can be approached using the same arguments as in Sect. 4, but is not explored further here. "

Additional experimentation in future work could explore this further.

4. Line 77 Instead of 'observation error covariance **R** diagonal' say 'diagonal observation error covariance **R**'. In every reference to **R**, the word matrix must be included.

Made the change and changed "variance" and "covariance" to "covariance matrix" including all references to R outside of equations.

5. Line 90 (Eq 1). The state vector is represented by letter $x(t)$ while before that is represented by the Greek letter $\chi(t)$. The notation must be consistent, all over the manuscript.

Modified text to clarify that $x$ refers to the ensemble while $\chi$ refers to the true state vector.

6. Eq 2 is assuming a direct observation vector y of the state vector $x$, i.e. an identity observation operator corresponding to the identity operator. The authors must say that explicitly in order to well understand the difference between $x$ and $y$. Moreover in eq. 2 and 3, a linear correction of the state vector due to time errors is assumed. Justify this approximation. The linearity hypothesis is only valid for small enough time errors or slow enough dynamics (small time derivative of the state vector). The smallness of time errors is not guaranteed for every type of observations, for instance when observation time is quite fuzzy (e.g. qualitative observations issued from old data, logbooks and diaries).

Text has been added immediately after Eq. 2 noting that this is an identity observation operator. The reviewer is correct that the linearity assumption is not guaranteed to be appropriate, nor can it be justified a priori. Text has been added at the end of Sect. 3.1 to note that it may not be appropriate, in which case the methods that involve interpolating between prior estimates of the model state at multiple times are available in the next subsection. The validity of the linear approximation is only readily assessed by exploring comparative results on a case-by-case basis. This is done in the results section for the highly idealized example model. Real applications would require additional care to see if the linear approximation resulted in increased error for specific applications.

7. Eq. 6 Use the consistent notation $\tau^2$ instead of τ^2.

Fixed.

8. Explain the title of section 4.3 'Impossible linear estimate' in this section, not in the following one.

Text was added at the start of Sect, 4.3 indicating why this method cannot be used in a real application, motivating the use of 'impossible'. Added a reference to the next section's discussion of a possible version of the method.

9. Line 140. Define the matrix Σ within the text.

Done.

10. Line 150 In eq. 13 the vector d must be used instead of d-tilde.

Fixed.

11. Line 153. Authors say, 'It is statistically difficult to distinguish between observation error and error due to time offset' Again this is only valid when linearity hypothesis is assumed leading to the additivity of observation error variance with the error variance coming from the time offset. Say that in the text.

We added a clause stating that this occurs when the linearity assumption is made.

12.1 Line 170. Present the reason for the choice of the '40-variable Lorenz model'. It is worth to present the model equations here:

dXi dt = Xi-1(Xi+1 − Xi−2) − Xi + F; i = 1,...,40 , X0 ≡ X40 , X-1 ≡ X39,

where the index is the positional index in a periodic ring.

Added the model equations. Added text indicating that Lorenz and Emmanuel chose this model for DA experimentation because of its size and convenient chaotic dynamics and referenced their work. Noted that this model has been used in a number of other DA studies (see next comment).

12.2 Justify the use of parameter F=8, maybe due to the existence of a chaotic attractor. Present other works using the same model in DA experiments:

*Evenson, G. and Fario, N.: Solving for the generalized inverse of the Lorenz model, J. Meteorol. Soc. Jpn., 75, 229–243, 1997.*

*van Leeuwen, P. J.: Nonlinear data assimilation in geosciences: an extremely efficient particle filter, Q. J. Roy. Meteor. Soc., 136, 1991–1999, 2010*

and nonlinear statistics like:

*Pires, C.A.L. & Perdigão, R.A.P. (2015) Non-Gaussian interaction information: estimation, optimization and diagnostic application of triadic wave resonance. Nonlinear Processes in Geophysics, 22, 87-108. DOI:10.5194/npg-22-87-2015*

Added a sentence explaining the choice of F=8 by Lorenz and Emmanuel and subsequent DA users. Added several references using L96 (and all of these use F=8 as the base choice).

13. Line 192. The random draws of the time offset are not allowed to be larger than PΔt in absolute value. In that case, authors halve artificially the error to 1/2PΔt which cuts the occurrence of large time offsets. The most correct procedure would be to take a truncated normal distribution (https://en.wikipedia.org/wiki/Truncated_normal_distribution). Authors must comment on that.

The reviewer is correct. We have rerun all the experiments with a truncated normal distribution as suggested. The number of instances where the original cases exceeded the bounds was small. The new results are quantitatively different for the cases with larger time error standard deviations, but qualitatively similar to the previous version. The revised code, results, and figures are now in the referenced repository. The text was modified to indicate this change.

14. Line 205. Authors say: 'This is analogous to the extrapolation performed by Eq. (2) using the ensemble mean estimate as a base point and gives a feeling for how nonlinear the time offset problem is at a particular analysis time.' Explain that describing the behavior in particular times (e.g. near time=369.6 the tangent linear hypothesis has completely failed).

Added additional sentences clarifying this and explicitly including the suggested example for when the linear assumption has failed.

15. Caption of Figure 1 must be completely revised. Refereed symbols do not correspond to the Table inserted in Figure 1 (e.g. green symbols do not exist).

We apologize for this unfortunate editing error that occurred when we modified the colors for a broader readership. This has been corrected.

16. In eqs. 17 and 18 do not use the right curly bracket }, since, mathematically, {} is used for the description of a set of members.

Replaced curly braces by cases environment.

17. Lines 255-259. The definition of $\boldsymbol{d}_i^m$ is quite confusing. Accordingly to the text, m is the positional index of the state vector. In lines 190-191, one says that all state variables are observed. Therefore, the observation vector has the same dimension than state vector and therefore the index i in $\boldsymbol{d}_i^m$ (The subscript $i$ references the vector component for a given scalar observation) is quite confusing. Explain that much better, using different indices for different purposes.

Thanks for pointing out this notational challenge. We improved the notation by replacing $d^m$ by $d^{\approx m}$ to make it clear that the superscript does not denote a vector index. We added text noting this and clarifying the notation.

18. Lines 260-265. The choice of the threshold T appears to be quite artificial and not generalizable to a generic model.

We agree and added an additional sentence noting this and indicating that tuning would be required for any real applications. For now, we have no insight on how to automate the selection of T.

19. In the caption of Fig. 2, say that the RMSE in each of the 10 experiments (IC trials), correspond to a dot in the graphs, using a particular DA method (color), analysis period and time offset std.

The caption has been modified accordingly.

**Reviewer 2:**

Major comments:

1. This paper makes numerous assumptions which are not clearly stated. Please go through the manuscript and clearly state all assumptions. Some examples are described below (and also in reviewer #1's comments).

We apologize for not being more comprehensive in the original submission. While the reviewer did not provide examples here, we tried to identify all missing assumptions and we dealt with all specific examples pointed out below by this reviewer and by Reviewer 1. Here is a list of assumptions now discussed in the paper (with line numbers from the revised manuscript):

- L 79: $\varepsilon_k^t$ is normally distributed, usually with mean 0.
- L 92: the observation operator is assumed to be the identity.
- L 95: $\Delta t$ is small enough that $\chi(t)$ is nearly linear at the scale of one timestep. This assumption is used to justify linear interpolation to get the state at intermediate times.
- L 120, 146, 171, 185: time errors are small enough that the system is locally linear at that scale.
- L 122: the average time derivative of the ensemble is a good estimate of the time derivative of the true state.
- L 192: The normal prior distribution with mean and variance given by the ensemble is a good estimate of the true state.

2. The notation in this manuscript is at times inconsistent and not all notation is defined. Again, some specifics are given below, but I have not documented all cases.

We have done our best to identify and correct notational issues. As an example, the use of $\varepsilon_k^t$ and $\tau$. $\tau$ was used both as a random variable ($\varepsilon_k^t$ conditioned on the observation) and as an input to the conditional PDF $p(\tau)$ in Sections 4.3 and 4.4. We renamed the random variable $\tau$ to $\varepsilon_k^t \mid y_k^o$ to fix this issue.

3. This one is a bit more open-ended. In this manuscript you incorporate time errors through a correction to the ensemble mean and variance of the prior estimate of the observations. Could you instead update your ensemble members directly (rather than their summary statistics)? For example, instead of using y_n = h(x_n(t_i)) could you use y_n = h(x_n(t_i + e_i,n)) where e_i,n is drawn from a distribution of time errors? How would this compare to your current method in terms of computational cost and ease of implementation?

This is a very interesting question and is related to a number of recent discussions in ensemble filtering independent of this particular time error problem. One can select different points in the EnKF algorithms to switch from a sample to a continuous approximation and back. When the problem is linear and Gaussian, there is generally no difference. As soon as those assumptions are discarded (like they should be for all real applications) the choice can qualitatively influence the results. Given that the paper is already way past the recommended length for NPG, we have chosen not to add an exploration here but it should be an interesting part of future work.

Minor comments:

— L 75: Why is it helpful to think of t^a_k as being equal to t_k * P? I agree this is true (assuming t_0=t^a_0=0, which is a reasonable), but the statement is confusing to me.

We concur and it has been removed.

— L 77: "where linear interpolation is used to compute \chi between the discrete times". I would rephrase this. You may use linear interpolation to compute \chi at non-integer-multiples of the time step, but this an approximation (unless the dynamics are linear). Something like the following would be more correct: "The kth observation is y_k \sim N(\chi(t_k^o), R). In general \chi need not be linear and since \chi is modeled only at discrete time steps we do not necessarily know \chi exactly at any non-integer multiple of the time step. In this study we make the assumption that the time steps are small enough so that the dynamics are approximately linear between two adjacent time steps. Note that without this assumption the performance of an ensemble Kalman filter may not be very good anyway. In practice, we use linear interpolation to compute \chi between the discrete times {t_i}."

We agree that the original text was inaccurately phrased. It has been revised to accurately describe how the forward operator is computed using a linearly interpolated approximation to the true state. We also included your point that if this assumption is violated, ensemble Kalman filters are going to struggle even without the time error issue.

— L 77: I would explicitly state that your forward operator/observation operator is the identity. (Also noted by reviewer #1).

Done.

— L 81: The notation for time offset here is different from the notation in L76. I suggest making the notation consistent.

In the original version, the actual confusion was in the use of the term 'time offset' to refer to two different things, rather than having two different notations refer to the same thing. This has been clarified and the notation was further cleaned up in the response to your major comment 2.

— L 83: I suggest defining "ensemble prior estimate of observations" sooner since it may not be clear to all readers. I see that you define it in equation (2).

Added a definition in parentheses.

— L 110: Is \tau known, or is the distribution of \tau known?

Replaced "an estimate of time offset" with "an estimate of the distribution of time offset".

— L 120: Please explain the equation for the distance.

Added "Assuming the system has locally linear behavior near time $t_k^a$, if $\varepsilon_k^t$ is sufficiently small we can approximate $\chi(t_k^o) = \chi(t_k^a + \varepsilon_k^t)$ as $\chi(t_k^a) + v\varepsilon_k^t$ . In that case, the difference between the observation and the truth is approximately $\tilde{d} = (y_k^o - \chi(t_k^o)) + (\chi(t_k^o) - \chi(t_k^a)) = \varepsilon_k^x + v\varepsilon_k^t$."

— Eq. (4): Define the variables used in this equation. I think it is worth writing out what you are doing here in a few steps (you can condense some of the algebra below if you need more space). As I see it, you are looking for the conditional probability that \epsilon^t = \tau given that the

difference (which is a random variable itself, call it D) is equal to d. From conditional probability this is proportional to the probability that \epsilon^t = \tau and D=d, or equivalently, \epsilon^x = d - \tau \nu. What you have here is somewhat confusing because \epsilon^t and \epsilon^x are independent, but conditioned on D=d they are not at all independent.

Added additional discussion of this equation and defined variables.

— Eq. (5): Reminder the reader that you are assuming a normal distribution.

Added "Since $\varepsilon_k^x$ and $\varepsilon_k^t$ are independent, the assumption that $\varepsilon_k^x$ and $\varepsilon_k^t$ are normally distributed gives…"

— L 129: If I understand correctly you introduce this term so that you can complete the square and simplify the expression. The word "absorbed" is used in different contexts here and in L 126. Consider using a different word here.

Changed to "Now, by introducing the constant factor […], we get the following without affecting the proportionality.

— L 141: This assumption is tricky with the time offset, but perhaps it is okay with your assumption of linearity between two adjacent time steps?

Agreed. It's tricky even in ensemble applications without the time offset. The text notes that for any real application we know there are going to be systematic differences from the real distribution, and the time error won't do anything but make that worse. The fact that the algorithm improves over doing nothing suggests that the violation isn't too egregious.

— L 146: Observation error generally means the difference between the observation and the truth, but that is not how I understand \epsilon^p. Please explain.

Made it clearer that $\varepsilon_k^x$ is observation error and $\varepsilon_k^p$ is a random draw from the shifted-over prior distribution.

— Eq. (12): See comment about Eq. (4)

As in the revision for Eq. 4, replaced $P\left(\varepsilon_k^\delta = \boldsymbol{\epsilon} \cap \varepsilon_k^t = \tau \mid \boldsymbol{\epsilon} + \tau\boldsymbol{v} = \boldsymbol{d}\right)$ by $P\left(\varepsilon_k^t = \tau \mid \varepsilon_k^\delta + \boldsymbol{v}\varepsilon_k^t = \boldsymbol{d}\right)$.

— Eq. (15): I don't follow this equation. Please explain and be specific about your notation.

We greatly expanded the level of detail and hopefully the clarity in all the derivations in this section as well as adding the generalization consistent with Reviewer 1's comment 3.

— L 171: Why do you make the choice to use a smaller timestep?

Added the following: "The choice to use a smaller timestep is intended to make the timestep smaller than the values of $\sigma_t$ for which the algorithm was tested. If $\Delta t$ were much larger than $\sigma_t$, then most true observation times would be within one timestep of the reported observation times. Since linear interpolation was used to compute the states of the system between timesteps, this would lead to time error contributing a linear factor to overall observation error. In practical

applications, we are quite interested in the effect of the system's nonlinearity on the total error in the presence of time error, which would not be represented in the experiment if the timestep were larger."

— L 175: Is 100 time steps enough to reach a statistically steady state?

Added clarification that we are discarding 100 analysis steps (500-6000 timesteps) and not 100 timesteps. Added text that empirical examination of RMSE time series suggests that things have equilibrated well before 100 steps in all cases.

— Eq. (17): "." Is used instead of ","

Fixed.

— Fig. 1: Check colors for consistency with text. Also check that they are colorblind-friendly.

Fixed colors in text to be consistent with figure. We've checked the colorblind-friendliness for this a few times.

---

## Author Response (AR2)

**Reviewer 1:**

1. Add the previously suggested reference:

Pires, C.A.L. & Perdigão, R.A.P. (2015) Non-Gaussian interaction information: estimation, optimization and diagnostic application of triadic wave resonance. Nonlinear Processes in Geophysics, 22, 87-108. DOI:10.5194/npg-22-87-2015

when authors present applications of (Lorenz and Emanuel 1998) model, also used in the manuscript.

This reference has now been added as suggested.